# On Multiplicative Integration with Recurrent Neural Networks

**Yuhuai Wu**[1,*], **Saizheng Zhang**[2,*], **Ying Zhang**[2], **Yoshua Bengio**[2,4] **and Ruslan Salakhutdinov**[3,4]
[1]University of Toronto, [2]MILA, Université de Montréal, [3]Carnegie Mellon University, [4]CIFAR
ywu@cs.toronto.edu,[2]{firstname.lastname}@umontreal.ca,rsalakhu@cs.cmu.edu

## Abstract

We introduce a general and simple structural design called "Multiplicative Integration" (MI) to improve recurrent neural networks (RNNs). MI changes the way in which information from difference sources flows and is integrated in the computational building block of an RNN, while introducing almost no extra parameters. The new structure can be easily embedded into many popular RNN models, including LSTMs and GRUs. We empirically analyze its learning behaviour and conduct evaluations on several tasks using different RNN models. Our experimental results demonstrate that Multiplicative Integration can provide a substantial performance boost over many of the existing RNN models.

## 1 Introduction

Recently there has been a resurgence of new structural designs for recurrent neural networks (RNNs) [1, 2, 3]. Most of these designs are derived from popular structures including vanilla RNNs, Long Short Term Memory networks (LSTMs) [4] and Gated Recurrent Units (GRUs) [5]. Despite of their varying characteristics, most of them share a common computational building block, described by the following equation:

$$\phi(\mathbf{W}\boldsymbol{x} + \mathbf{U}\boldsymbol{z} + \mathbf{b}), \tag{1}$$

where $\boldsymbol{x} \in \mathbb{R}^n$ and $\boldsymbol{z} \in \mathbb{R}^m$ are state vectors coming from different information sources, $\mathbf{W} \in \mathbb{R}^{d \times n}$ and $\mathbf{U} \in \mathbb{R}^{d \times m}$ are state-to-state transition matrices, and $\boldsymbol{b}$ is a bias vector. This computational building block serves as a combinator for integrating information flow from the $\boldsymbol{x}$ and $\boldsymbol{z}$ by a sum operation "+", followed by a nonlinearity $\phi$. We refer to it as the *additive building block*. Additive building blocks are widely implemented in various state computations in RNNs (e.g. hidden state computations for vanilla-RNNs, gate/cell computations of LSTMs and GRUs.

In this work, we propose an alternative design for constructing the computational building block by changing the procedure of information integration. Specifically, instead of utilizing sum operation "+", we propose to use the Hadamard product "⊙" to fuse $\mathbf{W}\boldsymbol{x}$ and $\mathbf{U}\boldsymbol{z}$:

$$\phi(\mathbf{W}\boldsymbol{x} \odot \mathbf{U}\boldsymbol{z} + \mathbf{b}) \tag{2}$$

The result of this modification changes the RNN from first order to second order [6], while introducing no extra parameters. We call this kind of information integration design a form of *Multiplicative Integration*. The effect of multiplication naturally results in a gating type structure, in which $\mathbf{W}\boldsymbol{x}$ and $\mathbf{U}\boldsymbol{z}$ are the gates of each other. More specifically, one can think of the state-to-state computation $\mathbf{U}\boldsymbol{z}$ (where for example $\boldsymbol{z}$ represents the previous state) as dynamically rescaled by $\mathbf{W}\boldsymbol{x}$ (where for example $\boldsymbol{x}$ represents the input). Such rescaling does not exist in the additive building block, in which $\mathbf{U}\boldsymbol{z}$ is independent of $\boldsymbol{x}$. This relatively simple modification brings about advantages over the additive building block as it alters RNN's gradient properties, which we discuss in detail in the next section, as well as verify through extensive experiments.

---

[*]Equal contribution.

In the following sections, we first introduce a general formulation of Multiplicative Integration. We then compare it to the additive building block on several sequence learning tasks, including character level language modelling, speech recognition, large scale sentence representation learning using a Skip-Thought model, and teaching a machine to read and comprehend for a question answering task. The experimental results (together with several existing state-of-the-art models) show that various RNN structures (including vanilla RNNs, LSTMs, and GRUs) equipped with Multiplicative Integration provide better generalization and easier optimization. Its main advantages include: (1) it enjoys better gradient properties due to the gating effect. Most of the hidden units are non-saturated; (2) the general formulation of Multiplicative Integration naturally includes the regular additive building block as a special case, and introduces almost no extra parameters compared to the additive building block; and (3) it is a drop-in replacement for the additive building block in most of the popular RNN models, including LSTMs and GRUs. It can also be combined with other RNN training techniques such as Recurrent Batch Normalization [7]. We further discuss its relationship to existing models, including Hidden Markov Models (HMMs) [8], second order RNNs [6] and Multiplicative RNNs [9].

## 2 Structure Description and Analysis

### 2.1 General Formulation of Multiplicative Integration

The key idea behind Multiplicative Integration is to integrate different information flows $\mathbf{W}\boldsymbol{x}$ and $\mathbf{U}\boldsymbol{z}$, by the Hadamard product "$\odot$". A more general formulation of Multiplicative Integration includes two more bias vectors $\boldsymbol{\beta}_1$ and $\boldsymbol{\beta}_2$ added to $\mathbf{W}\boldsymbol{x}$ and $\mathbf{U}\boldsymbol{z}$:

$$\phi((\mathbf{W}\boldsymbol{x} + \boldsymbol{\beta}_1) \odot (\mathbf{U}\boldsymbol{z} + \boldsymbol{\beta}_2) + \boldsymbol{b}) \tag{3}$$

where $\boldsymbol{\beta}_1, \boldsymbol{\beta}_2 \in \mathbb{R}^d$ are bias vectors. Notice that such formulation contains the first order terms as in a additive building block, i.e., $\boldsymbol{\beta}_1 \odot \mathbf{U}\boldsymbol{h}_{t-1} + \boldsymbol{\beta}_2 \odot \mathbf{W}\boldsymbol{x}_t$. In order to make the Multiplicative Integration more flexible, we introduce another bias vector $\boldsymbol{\alpha} \in \mathbb{R}^d$ to gate[2] the term $\mathbf{W}\boldsymbol{x} \odot \mathbf{U}\boldsymbol{z}$, obtaining the following formulation:

$$\phi(\boldsymbol{\alpha} \odot \mathbf{W}\boldsymbol{x} \odot \mathbf{U}\boldsymbol{z} + \boldsymbol{\beta}_1 \odot \mathbf{U}\boldsymbol{z} + \boldsymbol{\beta}_2 \odot \mathbf{W}\boldsymbol{x} + \boldsymbol{b}), \tag{4}$$

Note that the number of parameters of the Multiplicative Integration is about the same as that of the additive building block, since the number of new parameters ($\boldsymbol{\alpha}$, $\boldsymbol{\beta}_1$ and $\boldsymbol{\beta}_2$) are negligible compared to total number of parameters. Also, Multiplicative Integration can be easily extended to LSTMs and GRUs[3], that adopt vanilla building blocks for computing gates and output states, where one can directly replace them with the Multiplicative Integration. More generally, in any kind of structure where $k$ information flows ($k \geq 2$) are involved (e.g. residual networks [10]), one can implement pairwise Multiplicative Integration for integrating all $k$ information sources.

### 2.2 Gradient Properties

The Multiplicative Integration has different gradient properties compared to the additive building block. For clarity of presentation, we first look at vanilla-RNN and RNN with Multiplicative Integration embedded, referred to as **MI-RNN**. That is, $\boldsymbol{h}_t = \phi(\mathbf{W}\boldsymbol{x}_t + \mathbf{U}\boldsymbol{h}_{t-1} + \mathbf{b})$ versus $\boldsymbol{h}_t = \phi(\mathbf{W}\boldsymbol{x}_t \odot \mathbf{U}\boldsymbol{h}_{t-1} + \mathbf{b})$. In a vanilla-RNN, the gradient $\frac{\partial \boldsymbol{h}_t}{\partial \boldsymbol{h}_{t-n}}$ can be computed as follows:

$$\frac{\partial \boldsymbol{h}_t}{\partial \boldsymbol{h}_{t-n}} = \prod_{k=t-n+1}^{t} \mathbf{U}^T \mathrm{diag}(\phi'_k), \tag{5}$$

where $\phi'_k = \phi'(\mathbf{W}\boldsymbol{x}_k + \mathbf{U}\boldsymbol{h}_{k-1} + \mathbf{b})$. The equation above shows that the gradient flow through time heavily depends on the hidden-to-hidden matrix $\mathbf{U}$, but $\mathbf{W}$ and $\boldsymbol{x}_k$ appear to play a limited role: they only come in the derivative of $\phi'$ mixed with $\mathbf{U}\boldsymbol{h}_{k-1}$. On the other hand, the gradient $\frac{\partial \boldsymbol{h}_t}{\partial \boldsymbol{h}_{t-n}}$ of a MI-RNN is[4]:

$$\frac{\partial \boldsymbol{h}_t}{\partial \boldsymbol{h}_{t-n}} = \prod_{k=t-n+1}^{t} \mathbf{U}^T \mathrm{diag}(\mathbf{W}\boldsymbol{x}_k) \mathrm{diag}(\phi'_k), \tag{6}$$

where $\phi'_k = \phi'(\mathbf{W}\boldsymbol{x}_k \odot \mathbf{U}\boldsymbol{h}_{k-1} + \mathbf{b})$. By looking at the gradient, we see that the matrix $\mathbf{W}$ and the current input $\boldsymbol{x}_k$ is directly involved in the gradient computation by gating the matrix $\mathbf{U}$, hence more capable of altering the updates of the learning system. As we show in our experiments, with $\mathbf{W}\boldsymbol{x}_k$ directly gating the gradient, the vanishing/exploding problem is alleviated: $\mathbf{W}\boldsymbol{x}_k$ dynamically reconciles $\mathbf{U}$, making the gradient propagation easier compared to the regular RNNs. For LSTMs and GRUs with Multiplicative Integration, the gradient propagation properties are more complicated. But in principle, the benefits of the gating effect also persists in these models.

## 3 Experiments

In all of our experiments, we use the general form of Multiplicative Integration (Eq. 4) for any hidden state/gate computations, unless otherwise specified.

### 3.1 Exploratory Experiments

To further understand the functionality of Multiplicative Integration, we take a simple RNN for illustration, and perform several exploratory experiments on the character level language modeling task using Penn-Treebank dataset [11], following the data partition in [12]. The length of the training sequence is 50. All models have a single hidden layer of size 2048, and we use Adam optimization algorithm [13] with learning rate $1e^{-4}$. Weights are initialized to samples drawn from uniform$[-0.02, 0.02]$. Performance is evaluated by the bits-per-character (BPC) metric, which is $\log_2$ of perplexity.

#### 3.1.1 Gradient Properties

To analyze the gradient flow of the model, we divide the gradient in Eq. 6 into two parts: 1. the gated matrix products: $\mathbf{U}^T \mathrm{diag}(\mathbf{W}\boldsymbol{x}_k)$, and 2. the derivative of the nonlinearity $\phi'$, We separately analyze the properties of each term compared to the additive building block. We first focus on the gating effect brought by $\mathrm{diag}(\mathbf{W}\boldsymbol{x}_k)$. In order to separate out the effect of nonlinearity, we chose $\phi$ to be the identity map, hence both vanilla-RNN and MI-RNN reduce to linear models, referred to as **lin-RNN** and **lin-MI-RNN**.

For each model we monitor the log-L2-norm of the gradient $\log ||\partial C / \partial \boldsymbol{h}_t||_2$ (averaged over the training set) after every training epoch, where $\boldsymbol{h}_t$ is the hidden state at time step $t$, and $C$ is the negative log-likelihood of the single character prediction at the final time step ($t = 50$). Figure. 1 shows the evolution of the gradient norms for small $t$, i.e., $0, 5, 10$, as they better reflect the gradient propagation behaviour. Observe that the norms of lin-MI-RNN (orange) increase rapidly and soon exceed the corresponding norms of lin-RNN by a large margin. The norms of lin-RNN stay close to zero ($\approx 10^{-4}$) and their changes over time are almost negligible. This observation implies that with the help of $\mathrm{diag}(\mathbf{W}\boldsymbol{x}_k)$ term, the gradient vanishing of lin-MI-RNN can be alleviated compared to lin-RNN. The final test BPC (bits-per-character) of lin-MI-RNN is 1.48, which is comparable to a vanilla-RNN with stabilizing regularizer [14], while lin-RNN performs rather poorly, achieving a test BPC of over 2.

Next we look into the nonlinearity $\phi$. We chose $\phi = \tanh$ for both vanilla-RNN and MI-RNN. Figure 1 (c) and (d) shows a comparison of histograms of hidden activations over all time steps on the validation set after training. Interestingly, in (c) for vanilla-RNN, most activations are saturated with values around $\pm 1$, whereas in (d) for MI-RNN, most activations are non-saturated with values around 0. This has a direct consequence in gradient propagation: non-saturated activations imply that $\mathrm{diag}(\phi'_k) \approx 1$ for $\phi = tanh$, which can help gradients propagate, whereas saturated activations imply that $\mathrm{diag}(\phi'_k) \approx 0$, resulting in gradients vanishing.

#### 3.1.2 Scaling Problem

When adding two numbers at different order of magnitude, the smaller one might be negligible for the sum. However, when multiplying two numbers, the value of the product depends on both regardless of the scales. This principle also applies when comparing Multiplicative Integration to the additive building blocks. In this experiment, we test whether Multiplicative Integration is more robust to the scales of weight values. Following the same models as in Section 3.1.1, we first calculated the norms of $\mathbf{W}\boldsymbol{x}_k$ and $\mathbf{U}\boldsymbol{h}_{k-1}$ for both vanilla-RNN and MI-RNN for different $k$ after training. We found that in both structures, $\mathbf{W}\boldsymbol{x}_k$ is a lot smaller than $\mathbf{U}\boldsymbol{h}_{k-1}$ in magnitude. This might be due to the fact that $\boldsymbol{x}_k$ is a one-hot vector, making the number of updates for (columns of) $\mathbf{W}$ smaller than $\mathbf{U}$. As a result, in vanilla-RNN, the pre-activation term $\mathbf{W}\boldsymbol{x}_k + \mathbf{U}\boldsymbol{h}_{k-1}$ is largely controlled by the value of $\mathbf{U}\boldsymbol{h}_{k-1}$, while $\mathbf{W}\boldsymbol{x}_k$ becomes rather small. In MI-RNN, on the other hand, the pre-activation term $\mathbf{W}\boldsymbol{x}_k \odot \mathbf{U}\boldsymbol{h}_{k-1}$ still depends on the values of both $\mathbf{W}\boldsymbol{x}_k$ and $\mathbf{U}\boldsymbol{h}_{k-1}$, due to multiplication.

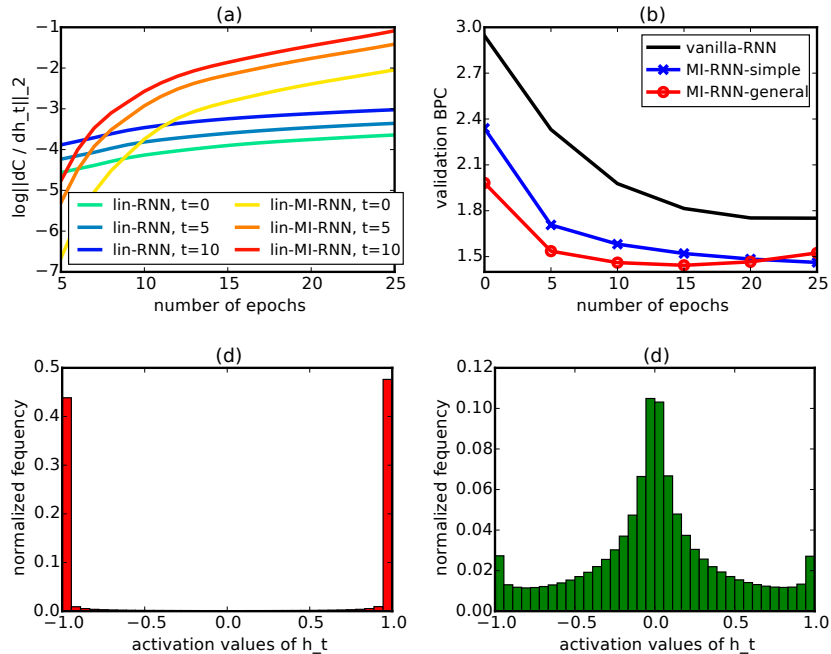

Figure 1: (a) Curves of log-L2-norm of gradients for lin-RNN (blue) and lin-MI-RNN (orange). Time gradually changes from {1, 5, 10}. (b) Validation BPC curves for vanilla-RNN, MI-RNN-simple using Eq. 2, and MI-RNN-general using Eq. 4. (c) Histogram of vanilla-RNN's hidden activations over the validation set, most activations are saturated. (d) Histogram of MI-RNN's hidden activations over the validation set, most activations are **not** saturated.

We next tried different initialization of $\mathbf{W}$ and $\mathbf{U}$ to test their sensitivities to the scaling. For each model, we fix the initialization of $\mathbf{U}$ to uniform$[-0.02, 0.02]$ and initialize $\mathbf{W}$ to uniform$[-r_{\mathbf{W}}, r_{\mathbf{W}}]$ where $r_{\mathbf{W}}$ varies in $\{0.02, 0.1, 0.3, 0.6\}$. Table 1, top left panel, shows results. As we increase the scale of $\mathbf{W}$, performance of the vanilla-RNN improves, suggesting that the model is able to better utilize the input information. On the other hand, MI-RNN is much more robust to different initializations, where the scaling has almost no effect on the final performance.

### 3.1.3 On different choices of the formulation

In our third experiment, we evaluated the performance of different computational building blocks, which are Eq. 1 (**vanilla-RNN**), Eq. 2 (**MI-RNN-simple**) and Eq. 4 (**MI-RNN-general**)[5]. From the validation curves in Figure 1 (b), we see that both MI-RNN, simple and MI-RNN-general yield much better performance compared to vanilla-RNN, and MI-RNN-general has a faster convergence speed compared to MI-RNN-simple. We also compared our results to the previously published models in Table 1, bottom left panel, where MI-RNN-general achieves a test BPC of 1.39, which is to our knowledge the best result for RNNs on this task without complex gating/cell mechanisms.

## 3.2 Character Level Language Modeling

In addition to the Penn-Treebank dataset, we also perform character level language modeling on two larger datasets: *text8*[6] and *Hutter Challenge Wikipedia*[7]. Both of them contain 100M characters from Wikipedia while *text8* has an alphabet size of 27 and *Hutter Challenge Wikipedia* has an alphabet size of 205. For both datasets, we follow the training protocols in [12] and [1] respectively. We use Adam for optimization with the starting learning rate grid-searched in $\{0.002, 0.001, 0.0005\}$. If the validation BPC (bits-per-character) does not decrease for 2 epochs, we half the learning rate.

We implemented Multiplicative Integration on both vanilla-RNN and LSTM, referred to as **MI-RNN** and **MI-LSTM**. The results for the *text8* dataset are shown in Table 1, bottom middle panel. All five models, including some of the previously published models, have the same number of

| $r_{\mathbf{W}} =$ | 0.02 | 0.1 | 0.3 | 0.6 | std |
|---|---|---|---|---|---|
| RNN | 1.69 | 1.65 | 1.57 | 1.54 | 0.06 |
| MI-RNN | **1.39** | 1.40 | 1.40 | 1.41 | 0.008 |

| WSJ Corpus | CER | WER |
|---|---|---|
| DRNN+CTCbeamsearch [15] | 10.0 | 14.1 |
| Encoder-Decoder [16] | 6.4 | 9.3 |
| LSTM+CTCbeamsearch [17] | 9.2 | 8.7 |
| Eesen [18] | - | **7.3** |
| LSTM+CTC+WFST (ours) | 6.5 | 8.7 |
| MI-LSTM+CTC+WFST (ours) | **6.0** | 8.2 |

| Penn-Treebank | BPC |
|---|---|
| RNN [12] | 1.42 |
| HF-MRNN [12] | 1.41 |
| RNN+stabalization [14] | 1.48 |
| MI-RNN (ours) | **1.39** |
| linear MI-RNN (ours) | 1.48 |

| text8 | BPC |
|---|---|
| RNN+smoothReLu [19] | 1.55 |
| HF-MRNN [12] | 1.54 |
| MI-RNN (ours) | **1.52** |
| LSTM (ours) | 1.51 |
| MI-LSTM(ours) | **1.44** |

| HutterWikipedia | BPC |
|---|---|
| stacked-LSTM [20] | 1.67 |
| GF-LSTM [1] | 1.58 |
| grid-LSTM [2] | 1.47 |
| MI-LSTM (ours) | **1.44** |

Table 1: Top: test BPCs and the standard deviation of models with different scales of weight initializations. Top right: test CERs and WERs on WSJ corpus. Bottom left: test BPCs on character level Penn-Treebank dataset. Bottom middle: test BPCs on character level text8 dataset. Bottom right: test BPCs on character level Hutter Prize Wikipedia dataset.

parameters ($\approx$4M). For RNNs without complex gating/cell mechanisms (the first three results), our MI-RNN (with $\{\alpha, \beta_1, \beta_2, \mathbf{b}\}$ initialized as $\{2, 0.5, 0.5, 0\}$) performs the best, our MI-LSTM (with $\{\alpha, \beta_1, \beta_2, \mathbf{b}\}$ initialized as $\{1, 0.5, 0.5, 0\}$) outperforms all other models by a large margin[8].

On *Hutter Challenge Wikipedia* dataset, we compare our MI-LSTM (single layer with 2048 unit, $\approx$17M, with $\{\alpha, \beta_1, \beta_2, \mathbf{b}\}$ initialized as $\{1, 1, 1, 0\}$) to the previous stacked LSTM (7 layers, $\approx$27M) [20], GF-LSTM (5 layers, $\approx$20M) [1], and grid-LSTM (6 layers, $\approx$17M) [2]. Table 1, bottom right panel, shows results. Despite the simple structure compared to the sophisticated connection designs in GF-LSTM and grid-LSTM, our MI-LSTM outperforms all other models and achieves the new state-of-the-art on this task.

### 3.3 Speech Recognition

We next evaluate our models on Wall Street Journal (WSJ) corpus (available as LDC corpus LDC93S6B and LDC94S13B), where we use the full 81 hour set "si284" for training, set "dev93" for validation and set "eval92" for test. We follow the same data preparation process and model setting as in [18], and we use 59 characters as the targets for the acoustic modelling. Decoding is done with the CTC [21] based weighted finite-state transducers (WFSTs) [22] as proposed by [18].

Our model (referred to as **MI-LSTM+CTC+WFST**) consists of 4 bidirectional MI-LSTM layers, each with 320 units for each direction. CTC is performed on top to resolve the alignment issue in speech transcription. For comparison, we also train a baseline model (referred to as **LSTM+CTC+WFST**) with the same size but using vanilla LSTM. Adam with learning rate 0.0001 is used for optimization and Gaussian weight noise with zero mean and 0.05 standard deviation is injected for regularization. We evaluate our models on the character error rate (CER) without language model and the word error rate (WER) with extended trigram language model.

Table 1, top right panel, shows that MI-LSTM+CTC+WFST achieves quite good results on both CER and WER compared to recent works, and it has a clear improvement over the baseline model. Note that we did not conduct a careful hyper-parameter search on this task, hence one could potentially obtain better results with better decoding schemes and regularization techniques.

### 3.4 Learning Skip-Thought Vectors

Next, we evaluate our Multiplicative Integration on the Skip-Thought model of [23]. Skip-Thought is an encoder-decoder model that attempts to learn generic, distributed sentence representations. The model produces sentence representation that are robust and perform well in practice, as it achieves excellent results across many different NLP tasks. The model was trained on the BookCorpus dataset that consists of 11,038 books with 74,004,228 sentences. Not surprisingly, a single pass through

| Semantic-Relatedness | $r$ | $\rho$ | **MSE** |
|---|---|---|---|
| uni-skip [23] | 0.8477 | 0.7780 | 0.2872 |
| bi-skip [23] | 0.8405 | 0.7696 | 0.2995 |
| combine-skip [23] | 0.8584 | 0.7916 | 0.2687 |
| uni-skip (ours) | 0.8436 | 0.7735 | 0.2946 |
| MI-uni-skip (ours) | **0.8588** | **0.7952** | **0.2679** |

| Paraphrase detection | **Acc** | **F1** |
|---|---|---|
| uni-skip [23] | 73.0 | 81.9 |
| bi-skip [23] | 71.2 | 81.2 |
| combine-skip [23] | 73.0 | 82.0 |
| uni-skip (ours) | **74.0** | 81.9 |
| MI-uni-skip (ours) | **74.0** | 82.1 |

| Classification | MR | CR | SUBJ | MPQA |
|---|---|---|---|---|
| uni-skip [23] | 75.5 | 79.3 | 92.1 | 86.9 |
| bi-skip [23] | 73.9 | 77.9 | 92.5 | 83.3 |
| combine-skip [23] | 76.5 | 80.1 | **93.6** | 87.1 |
| uni-skip (ours) | 75.9 | 80.1 | 93.0 | 87.0 |
| MI-uni-skip (ours) | **77.9** | **82.3** | 93.3 | **88.1** |

| Attentive Reader | Val. Err. |
|---|---|
| LSTM [7] | 0.5033 |
| BN-LSTM [7] | 0.4951 |
| BN-everywhere [7] | 0.5000 |
| LSTM (ours) | 0.5053 |
| MI-LSTM (ours) | 0.4721 |
| MI-LSTM+BN (ours) | 0.4685 |
| MI-LSTM+BN-everywhere (ours) | **0.4644** |

Table 2: Top left: skip-thought+MI on Semantic-Relatedness task. Top Right: skip-thought+MI on Paraphrase Detection task. Bottom left: skip-thought+MI on four different classification tasks. Bottom right: Multiplicative Integration (with batch normalization) on Teaching Machines to Read and Comprehend task.

the training data can take up to a week on a high-end GPU (as reported in [23]). Such training speed largely limits one to perform careful hyper-parameter search. However, with Multiplicative Integration, not only the training time is shortened by a factor of two, but the final performance is also significantly improved.

We exactly follow the authors' Theano implementation of the skip-thought model[9]: Encoder and decoder are single-layer GRUs with hidden-layer size of 2400; all recurrent matrices adopt orthogonal initialization while non-recurrent weights are initialized from uniform distribution. Adam is used for optimization. We implemented Multiplicative Integration only for the encoder GRU (embedding MI into decoder did not provide any substantial gains). We refer our model as **MI-uni-skip**, with $\{\boldsymbol{\alpha}, \boldsymbol{\beta}_1, \boldsymbol{\beta}_2, \mathbf{b}\}$ initialized as $\{1, 1, 1, 0\}$. We also train a baseline model with the same size, referred to as **uni-skip(ours)**, which essentially reproduces the original model of [23].

During the course of training, we evaluated the skip-thought vectors on the semantic relatedness task, using SICK dataset, every 2500 updates for both MI-uni-skip and the baseline model (each iteration processes a mini-batch of size 64). The results are shown in Figure 2a. Note that MI-uni-skip significantly outperforms the baseline, not only in terms of speed of convergence, but also in terms of final performance. At around 125k updates, MI-uni-skip already exceeds the best performance achieved by the baseline, which takes about twice the number of updates.

We also evaluated both models after one week of training, with the best results being reported on six out of eight tasks reported in [23]: semantic relatedness task on SICK dataset, paraphrase detection task on Microsoft Research Paraphrase Corpus, and four classification benchmarks: movie review sentiment (MR), customer product reviews (CR), subjectivity/objectivity classification (SUBJ), and opinion polarity (MPQA). We also compared our results with the results reported on three models in the original skip-thought paper: **uni-skip**, **bi-skip**, **combine-skip**. Uni-skip is the same model as our baseline, bi-skip is a bidirectional model of the same size, and combine-skip takes the concatenation of the vectors from uni-skip and bi-skip to form a 4800 dimension vector for task evaluation. Table 2 shows that MI-uni-skip dominates across all the tasks. Not only it achieves higher performance than the baseline model, but in many cases, it also outperforms the combine-skip model, which has twice the number of dimensions. Clearly, Multiplicative Integration provides a faster and better way to train a large-scale Skip-Thought model.

### 3.5 Teaching Machines to Read and Comprehend

In our last experiment, we show that the use of Multiplicative Integration can be combined with other techniques for training RNNs, and the advantages of using MI still persist. Recently, [7] introduced Recurrent Batch-Normalization. They evaluated their proposed technique on a uni-

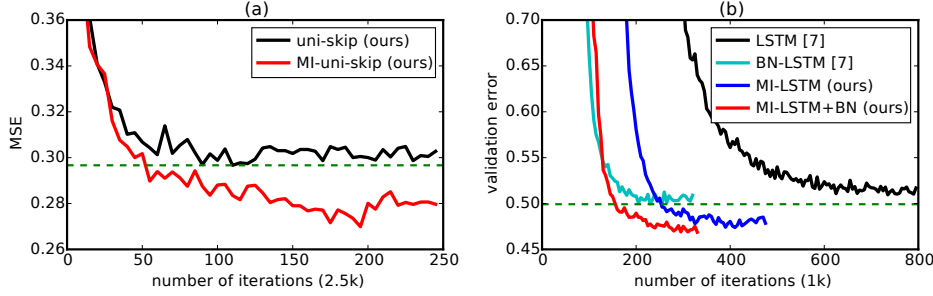

Figure 2: (a) MSE curves of uni-skip (ours) and MI-uni-skip (ours) on semantic relatedness task on SICK dataset. MI-uni-skip significantly outperforms baseline uni-skip. (b) Validation error curves on attentive reader models. There is a clear margin between models with and without MI.

directional Attentive Reader Model [24] for the question answering task using the CNN corpus[10]. To test our approach, we evaluated the following four models: 1. A vanilla LSTM attentive reader model with a single hidden layer size 240 (same as [7]) as our baseline, referred to as **LSTM (ours)**, 2. A multiplicative integration LSTM with a single hidden size 240, referred to as **MI-LSTM**, 3. MI-LSTM with Batch-Norm, referred to as **MI-LSTM+BN**, 4. MI-LSTM with Batch-Norm everywhere (as detailed in [7]), referred to as **MI-LSTM+BN-everywhere**. We compared our models to results reported in [7] (referred to as LSTM, BN-LSTM and BN-LSTM everywhere) [11].

For all MI models, $\{\boldsymbol{\alpha}, \boldsymbol{\beta}_1, \boldsymbol{\beta}_2, \mathbf{b}\}$ were initialized to $\{1, 1, 1, 0\}$. We follow the experimental protocol of [7][12] and use exactly the same settings as theirs, except we remove the gradient clipping for MI-LSTMs. Figure. 2b shows validation curves of the baseline (LSTM), MI-LSTM, BN-LSTM, and MI-LSTM+BN, and the final validation errors of all models are reported in Table 2, bottom right panel. Clearly, using Multiplicative Integration results in improved model performance regardless of whether Batch-Norm is used. However, the combination of MI and Batch-Norm provides the best performance and the fastest speed of convergence. This shows the general applicability of Multiplication Integration when combining it with other optimization techniques.

## 4 Relationship to Previous Models

### 4.1 Relationship to Hidden Markov Models

One can show that under certain constraints, MI-RNN is effectively implementing the forward algorithm of the Hidden Markov Model(HMM). A direct mapping can be constructed as follows (see [25] for a similar derivation). Let $\mathbf{U} \in \mathbb{R}^{m \times m}$ be the state transition probability matrix with $\mathbf{U}_{ij} = \Pr[h_{t+1} = i | h_t = j]$, $\mathbf{W} \in \mathbb{R}^{m \times n}$ be the observation probability matrix with $\mathbf{W}_{ij} = \Pr[x_t = i | h_t = j]$. When $\boldsymbol{x}_t$ is a one-hot vector (e.g., in many of the language modelling tasks), multiplying it by $\mathbf{W}$ is effectively choosing a column of the observation matrix. Namely, if the $j^{th}$ entry of $\boldsymbol{x}_t$ is one, then $\mathbf{W}\boldsymbol{x}_t = \Pr[x_t | h_t = j]$. Let $\boldsymbol{h}_0$ be the initial state distribution with $\boldsymbol{h}_0 = \Pr[h_0]$ and $\{\boldsymbol{h}_t\}_{t \geq 1}$ be the alpha values in the forward algorithm of HMM, i.e., $\boldsymbol{h}_t = \Pr[x_1, ..., x_t, h_t]$. Then $\mathbf{U}\boldsymbol{h}_t = \Pr[x_1, ..., x_t, h_{t+1}]$. Thus $\boldsymbol{h}_{t+1} = \mathbf{W}\boldsymbol{x}_{t+1} \odot \mathbf{U}\boldsymbol{h}_t = \Pr[x_{t+1} | h_{t+1}] \cdot \Pr[x_1, ..., x_t, h_{t+1}] = \Pr[x_1, ..., x_{t+1}, h_{t+1}]$. To exactly implement the forward algorithm using Multiplicative Integration, the matrices $\mathbf{W}$ and $\mathbf{U}$ have to be probability matrices, and $\boldsymbol{x}_t$ needs to be a one-hot vector. The function $\phi$ needs to be linear, and we drop all the bias terms. Therefore, RNN with Multiplicative Integration can be seen as a nonlinear extension of HMMs. The extra freedom in parameter values and nonlinearity makes the model more flexible compared to HMMs.

### 4.2 Relations to Second Order RNNs and Multiplicative RNNs

MI-RNN is related to the second order RNN [6] and the multiplicative RNN (MRNN) [9]. We first describe the similarities with these two models:

The second order RNN involves a second order term $\boldsymbol{s}_t$ in a vanilla-RNN, where the $i$th element $\boldsymbol{s}_{t,i}$ is computed by the bilinear form: $\boldsymbol{s}_{t,i} = \boldsymbol{x}_t^T \mathcal{T}^{(i)} \boldsymbol{h}_{t-1}$, where $\mathcal{T}^{(i)} \in \mathbb{R}^{n \times m} (1 \leq i \leq m)$ is

the $i$th slice of a tensor $\mathcal{T} \in \mathbb{R}^{m \times n \times m}$. Multiplicative Integration also involve a second order term $s_t = \boldsymbol{\alpha} \odot \mathbf{W} \boldsymbol{x}_t \odot \mathbf{U} \boldsymbol{h}_{t-1}$, but in our case $s_{t,i} = \alpha_i (\mathbf{w}_i \cdot \boldsymbol{x}_t)(\mathbf{u}_i \cdot \boldsymbol{h}_{t-1}) = \boldsymbol{x}_t^T (\alpha \mathbf{w}_i \otimes \mathbf{u}_i) \boldsymbol{h}_{t-1}$, where $\mathbf{w}_i$ and $\mathbf{u}_i$ are $i$th row in $\mathbf{W}$ and $\mathbf{U}$, and $\alpha_i$ is the $i$th element of $\boldsymbol{\alpha}$. Note that the outer product $\alpha_i \mathbf{w}_i \otimes \mathbf{u}_i$ is a rank-1 matrix. The Multiplicative RNN is also a second order RNN, but which approximates $\mathcal{T}$ by a tensor decomposition $\sum \boldsymbol{x}_t^{(i)} \mathcal{T}^{(i)} = \mathbf{P}\mathrm{diag}(\mathbf{V}\boldsymbol{x}_t)\mathbf{Q}$. For MI-RNN, we can also think of the second order term as a tensor decomposition: $\boldsymbol{\alpha} \odot \mathbf{W}\boldsymbol{x}_t \odot \mathbf{U}\boldsymbol{h}_{t-1} = \mathbf{U}(\boldsymbol{x}_t)\boldsymbol{h}_{t-1} = [\mathrm{diag}(\boldsymbol{\alpha})\mathrm{diag}(\mathbf{W}\boldsymbol{x}_t)\mathbf{U}]\boldsymbol{h}_{t-1}$.

There are however several differences that make MI a favourable model: (1) Simpler Parametrization: MI uses a rank-1 approximation compared to the second order RNNs, and a diagonal approximation compared to Multiplicative RNN. Moreover, MI-RNN shares parameters across the first and second order terms, whereas the other two models do not. As a result, the number of parameters are largely reduced, which makes our model more practical for large scale problems, while avoiding overfitting. (2) Easier Optimization: In tensor decomposition methods, the products of three different (low-rank) matrices generally makes it hard to optimize [9]. However, the optimization problem becomes easier in MI, as discussed in section 2 and 3. (3) General structural design vs. vanilla-RNN design: Multiplicative Integration can be easily embedded in many other RNN structures, e.g. LSTMs and GRUs, whereas the second order RNN and MRNN present a very specific design for modifying vanilla-RNNs.

Moreover, we also compared MI-RNN's performance to the previous HF-MRNN's results (Multiplicative RNN trained by Hessian-free method) in Table 1, bottom left and bottom middle panels, on Penn-Treebank and text8 datasets. One can see that MI-RNN outperforms HF-MRNN on both tasks.

### 4.3 General Multiplicative Integration

Multiplicative Integration can be viewed as a general way of combining information flows from two different sources. In particular, [26] proposed the ladder network that achieves promising results on semi-supervised learning. In their model, they combine the lateral connections and the backward connections via the "combinator" function by a Hadamard product. The performance would severely degrade without this product as empirically shown by [27]. [28] explored neural embedding approaches in knowledge bases by formulating relations as bilinear and/or linear mapping functions, and compared a variety of embedding models on the link prediction task. Surprisingly, the best results among all bilinear functions is the simple weighted Hadamard product. They further carefully compare the multiplicative and additive interactions and show that the multiplicative interaction dominates the additive one.

## 5 Conclusion

In this paper we proposed to use Multiplicative Integration (MI), a simple Hadamard product to combine information flow in recurrent neural networks. MI can be easily integrated into many popular RNN models, including LSTMs and GRUs, while introducing almost no extra parameters. Indeed, the implementation of MI requires almost no extra work beyond implementing RNN models. We also show that MI achieves state-of-the-art performance on four different tasks or 11 datasets of varying sizes and scales. We believe that the Multiplicative Integration can become a default building block for training various types of RNN models.

### Acknowledgments

The authors acknowledge the following agencies for funding and support: NSERC, Canada Research Chairs, CIFAR, Calcul Quebec, Compute Canada, Disney research and ONR Grant N000141310721. The authors thank the developers of Theano [29] and Keras [30], and also thank Jimmy Ba for many thought-provoking discussions.

## Footnotes

[2]If $\boldsymbol{\alpha} = \mathbf{0}$, the Multiplicative Integration will degenerate to the vanilla additive building block.

[3]See exact formulations in the Appendix.

[4]Here we adopt the simplest formulation of Multiplicative Integration for illustration. In the more general case (Eq. 4), $\mathrm{diag}(\mathbf{W}\boldsymbol{x}_k)$ in Eq. 6 will become $\mathrm{diag}(\boldsymbol{\alpha} \odot \mathbf{W}\boldsymbol{x}_k + \boldsymbol{\beta}_1)$.

[5]We perform hyper-parameter search for the initialization of $\{\boldsymbol{\alpha}, \boldsymbol{\beta}_1, \boldsymbol{\beta}_2, \mathbf{b}\}$ in MI-RNN-general.

[6]http://mattmahoney.net/dc/textdata

[7]http://prize.hutter1.net/

[8][7] reports better results but they use much larger models ($\approx$16M) which is not directly comparable.

[9] https://github.com/ryankiros/skip-thoughts

[10]Note that [7] used a truncated version of the original dataset in order to save computation.

[11]Learning curves and the final result number are obtained by emails correspondence with authors of [7].

[12]https://github.com/cooijmanstim/recurrent-batch-normalization.git.

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
