[Supplementary Material]

# Appendix

## A  Implementation Details

### A.1  MI-LSTM

Our MI-LSTM (without peephole connection) in experiments has the follow formulation:

$$\boldsymbol{z}_t = \tanh(\boldsymbol{\alpha}_z \odot \mathbf{W}_z \boldsymbol{x}_t \odot \mathbf{U}_z \boldsymbol{h}_{t-1} + \boldsymbol{\beta}_{z,1} \odot \mathbf{U}_z \boldsymbol{h}_{t-1} + \boldsymbol{\beta}_{z,2} \odot \mathbf{W}_z \boldsymbol{x}_t + \mathbf{b}_z) \qquad \textbf{block input}$$

$$\boldsymbol{i}_t = \sigma(\boldsymbol{\alpha}_i \odot \mathbf{W}_i \boldsymbol{x}_t \odot \mathbf{U}_i \boldsymbol{h}_{t-1} + \boldsymbol{\beta}_{i,1} \odot \mathbf{U}_i \boldsymbol{h}_{t-1} + \boldsymbol{\beta}_{i,2} \odot \mathbf{W}_i \boldsymbol{x}_t + \mathbf{b}_i) \qquad \textbf{input gate}$$

$$\boldsymbol{f}_t = \sigma(\boldsymbol{\alpha}_f \odot \mathbf{W}_f \boldsymbol{x}_t \odot \mathbf{U}_f \boldsymbol{h}_{t-1} + \boldsymbol{\beta}_{f,1} \odot \mathbf{U}_f \boldsymbol{h}_{t-1} + \boldsymbol{\beta}_{f,2} \odot \mathbf{W}_f \boldsymbol{x}_t + \mathbf{b}_f) \qquad \textbf{forget gate}$$

$$\boldsymbol{c}_t = \boldsymbol{i}_t \odot \boldsymbol{z}_t + \boldsymbol{f}_t \odot \boldsymbol{c}_{t-1} \qquad \textbf{cell state}$$

$$\boldsymbol{o}_t = \sigma(\boldsymbol{\alpha}_o \odot \mathbf{W}_o \boldsymbol{x}_t \odot \mathbf{U}_o \boldsymbol{h}_{t-1} + \boldsymbol{\beta}_{o,1} \odot \mathbf{U}_o \boldsymbol{h}_{t-1} + \boldsymbol{\beta}_{o,2} \odot \mathbf{W}_o \boldsymbol{x}_t + \mathbf{b}_o) \qquad \textbf{output gate}$$

$$\boldsymbol{h}_t = \boldsymbol{o}_t \odot \tanh(\boldsymbol{c}_t) \qquad \textbf{block output}$$

where $\{\boldsymbol{\alpha}_*, \boldsymbol{\beta}_{*,1}, \boldsymbol{\beta}_{*,2}\}_{*=z,i,f,o}$ are bias vectors, $\sigma$ denotes the sigmoid function.

### A.2  MI-GRU

Our MI-GRU in experiments has the follow formulation:

$$\boldsymbol{z}_t = \sigma(\boldsymbol{\alpha}_z \odot \mathbf{W}_z \boldsymbol{x}_t \odot \mathbf{U}_z \boldsymbol{h}_{t-1} + \boldsymbol{\beta}_{z,1} \odot \mathbf{U}_z \boldsymbol{h}_{t-1} + \boldsymbol{\beta}_{z,2} \odot \mathbf{W}_z \boldsymbol{x}_t + \mathbf{b}_z) \qquad \textbf{update gate}$$

$$\boldsymbol{r}_t = \sigma(\boldsymbol{\alpha}_r \odot \mathbf{W}_r \boldsymbol{x}_t \odot \mathbf{U}_r \boldsymbol{h}_{t-1} + \boldsymbol{\beta}_{r,1} \odot \mathbf{U}_r \boldsymbol{h}_{t-1} + \boldsymbol{\beta}_{r,2} \odot \mathbf{W}_r \boldsymbol{x}_t + \mathbf{b}_r) \qquad \textbf{reset gate}$$

$$\tilde{\boldsymbol{h}}_t = \tanh(\boldsymbol{\alpha}_h \odot \mathbf{W}_h \boldsymbol{x}_t \odot \mathbf{U}_h \left(\boldsymbol{r}_t \odot \boldsymbol{h}_{t-1}\right) + \boldsymbol{\beta}_{h,1} \odot \mathbf{U}_h \left(\boldsymbol{r}_t \odot \boldsymbol{h}_{t-1}\right) + \boldsymbol{\beta}_{h,2} \odot \mathbf{W}_h \boldsymbol{x}_t + \mathbf{b}_h)$$

$$\textbf{candidate activation}$$

$$\boldsymbol{h}_t = (1 - \odot \boldsymbol{z}_t) \odot \boldsymbol{h}_{t-1} + \boldsymbol{z}_t \odot \tilde{\boldsymbol{h}}_{t-1} \qquad \textbf{hidden activation}$$

where $\{\boldsymbol{\alpha}_*, \boldsymbol{\beta}_{*,1}, \boldsymbol{\beta}_{*,2}\}_{*=z,r,h}$ are bias vectors, $\sigma$ denotes the sigmoid function.