[Reviews · NeurIPS 2016]

Reviewer 1

Summary

This paper has a simple premise: that the, say, LSTM cell works better with multiplicative updates (equation 2) rather than additive ones (equation 1). This additive update is used in various places in lieu of additive ones, in various places in the LSTM recurrence equations (the exact formulation is in the supplementary material). A slightly hand wavy argument is made in favour of the multiplicative update, on the grounds of superior gradient flow (section 2.2). Mainly however, the authors make a rather thorough empirical investigation which shows remarkably good performance of their new architectures, on a range of real problems. Figure 1(a) is nice, showing an apparent greater information flow (as defined by a particular gradient) through time for the new scheme, as well as faster convergence and less saturated hidden unit activations. Overall, the experimental results appear thorough and convincing, although I am not a specialist in this area.

Qualitative Assessment

I have no major criticisms of the paper. I like the presentation, and the simplicity of the proposed scheme. However, I’m not a specialist in the area so not ideally placed to evaluate the potential impact of the method / importance of the experimental results.

Confidence in this Review

2-Confident (read it all; understood it all reasonably well)


Reviewer 2

Summary

This model presents a multiplicative alternative (with an additive component) to the additive update which happens at the core of various RNNs (Simple RNNs, GRUs, LSTMs). The multiplicative component, without introducing a significant change in the number of parameters, yields better gradient passing properties which enable the learning of better models, as shown in experiments.

Qualitative Assessment

My review will be short as there is not much to say. The paper is clearly written. The model changes are obvious and simple, which is the mark of a good idea. The explanation of the backward dynamics of these updates is trivial once explained, as is done clearly here. The experiments are diverse and convincing. I have no significant criticisms, as the authors points out the relation to multiplicative RNNs in section 4, although I would ideally like a bit more detail here if space allows it. Great work.

Confidence in this Review

3-Expert (read the paper in detail, know the area, quite certain of my opinion)


Reviewer 3

Summary

In this paper the authors advocate for the use of multiplicative interactions in RNNs. The sum between the projected hidden state and the input embedding is replaced by a component wise product. They show results on a large variety of tasks with small but consistent improvements. They also claim faster convergence.

Qualitative Assessment

My biggest concern about this work is the lack of novelty. Despite the claimed differences, the proposed method is a special case of what proposed in [10]. I doubt that the slight different parameterization (remove one factor-hidden matrix and introduce more bias terms) makes much difference. I strongly suspect that the improved performance is due to better optimization (HF has proven to be very brittle). I also found weak the argument for which gating makes gradients flow better because there is no guarantee this is going to happen. The gradient has an additional term it's multiplied by, There is no guarantee that the Jacobian between two consecutive hidden states has unitary eigenvalues... The experimental section is solid, although improvements are rather marginal yet consistent. -------------------------------------------------- I have read the rebuttal. I sincerely apologize with the authors. There is no fatal flaw in their paper, I just clicked that by mistake. Apologies. I still think that the paper lacks novelty, however. [10] proposes multiplicative interactions for a plain RNN but there is no reason why the same would not apply to other types of RNNs like LSTMs. Multiplicative interactions apply to anything really (true for [10] and this paper). In [10] they had: h_t = non_linearity( V ((W_x x) .* (W_h h_{t-1})) ) here the authors propose: h_t = non_linearity( \alpha .* ((W_x x) .* (W_h h_{t-1})) + bias_terms) I do understand that this is a simplification of [10], but is it so different? The difference is removing matrix V above and introducing bias terms. If they authors acknowledged more and analyzed the effect of each of these modifications, it would have been more informative in my opinion. That said, this paper has pretty solid empirical validation which compensates the lack of novelty. Based on this, I'd certainly recommend acceptance, but I would be quite hesitant about promoting this to an oral.

Confidence in this Review

2-Confident (read it all; understood it all reasonably well)


Reviewer 4

Summary

The paper proposes to use component wise multiplication to integrate two streams of information in a neural architecture, specifically to integrate the input stream and the hidden state stream in recurrent neural networks. The paper reports a wide range of good results giving credence to the multiplicative integration idea.

Qualitative Assessment

The paper is well written. The idea is sound and well argued for. The set of experiments is also quite strong - although none of the gaps with previous methods is that remarkable. It would have been nice to see a (real world) task where multiplicative integration would give a very large jump in performance, for instance, like the one seen for residual networks. But overall, the contributions in this paper are significant.

Confidence in this Review

2-Confident (read it all; understood it all reasonably well)